# Combination of Radiomics and Machine Learning with Diffusion-Weighted MR Imaging for Clinical Outcome Prognostication in Cervical Cancer

Ankush Jajodia [1,*], Ayushi Gupta [2], Helmut Prosch [3], Marius Mayerhoefer [4], Swarupa Mitra [5], Sunil Pasricha [6], Anurag Mehta [7], Sunil Puri [1] and Arvind Chaturvedi [1]

1 Department of Radiology, Rajiv Gandhi Cancer Institute and Research Centre, New Delhi 110085, India; skpurigbph@yahoo.co.in (S.P.); arvind.chaturvedi@gmail.com (A.C.)
2 Center for Computational Biology, Indraprastha Institute of Information Technology, New Delhi 110020, India; ayushi16123@iiitd.ac.in
3 Department of Biomedical Imaging and Image-Guided Therapy, Medical University of Vienna, 1090 Vienna, Austria; helmut.prosch@meduniwien.ac.at
4 Department of Radiology, Memorial Sloan Kettering Cancer Center, New York, NY 10065, USA; marius.mayerhoefer@meduniwien.ac.at
5 Department of Radiation Oncology, Rajiv Gandhi Cancer Institute and Research Centre, New Delhi 110085, India; swarupamitra@gmail.com
6 Department of Laboratory & Histopathology, Rajiv Gandhi Cancer Institute, New Delhi 110085, India; drsunilpasricha@yahoo.com
7 Department of Laboratory & Transfusion Services and Director Research, Rajiv Gandhi Cancer Institute, New Delhi 110085, India; anumehta11@gmail.com
* Correspondence: ankushjaj@gmail.com

**Abstract:** Objectives: To explore the potential of Radiomics alone and in combination with a diffusion-weighted derived quantitative parameter, namely the apparent diffusion co-efficient (ADC), using supervised classification algorithms in the prediction of outcomes and prognosis. Materials and Methods: Retrospective evaluation of the imaging was conducted for a study cohort of uterine cervical cancer, candidates for radical treatment with chemo radiation. ADC values were calculated from the darkest part of the tumor, both before (labeled preADC) and post treatment (labeled postADC) with chemo radiation. Post extraction of 851 Radiomics features and feature selection analysis—by taking the union of the features that had Pearson correlation >0.35 for recurrence, >0.49 for lymph node and >0.40 for metastasis—was performed to predict clinical outcomes. Results: The study enrolled 52 patients who presented with variable FIGO stages in the age range of 28–79 (Median = 53 years) with a median follow-up of 26.5 months (range: 7–76 months). Disease recurrence occurred in 12 patients (23%). Metastasis occurred in 15 patients (28%). A model generated with 24 radiomics features and preADC using a monotone multi-layer perceptron neural network to predict the recurrence yields an AUC of 0.80 and a Kappa value of 0.55 and shows that the addition of radiomics features to ADC values improves the statistical metrics by approximately 40% for AUC and approximately 223% for Kappa. Similarly, the neural network model for prediction of metastasis returns an AUC value of 0.84 and a Kappa value of 0.65, thus exceeding performance expectations by approximately 25% for AUC and approximately 140% for Kappa. There was a significant input of GLSZM features (SALGLE and LGLZE) and GLDM features (SDLGLE and DE) in correlation with clinical outcomes of recurrence and metastasis. Conclusions: The study is an effort to bridge the unmet need of translational predictive biomarkers in the stratification of uterine cervical cancer patients based on prognosis.

**Keywords:** radiomics; diffusion-weighted; MRI; cervical cancer

## 1. Introduction

Uterine cervical cancer incidence ranks seventh of all cancers [1] and represents a significant burden in low and middle-income nations [2], while in developing nations, it accounts for leading mortality in women [3]. Radiation with concurrent chemotherapy forms the cornerstone of the management of stage IB2 to IVA uterine cervical cancer disease for the International Federation of Gynecology and Obstetrics (FIGO) [4]. This treatment strategy, used in neoadjuvant settings, has gained popularity because of excellent responses leading to the debulking of these tumors, and further, leading to shrinkage and possible resection [5–7]. Tumor size, histology, lymph node involvement, the infiltration of neighboring structures and the presence of distant metastases are steering the prognostication in cervical cancer [8], with a notable discrepancy in the prognosis among patients belonging to the same stage, which could not be attributed to the constellation of clinico-pathological features [9]. The determination of a few of these factors requires representative tumor tissue, which warrants invasive procedures that add to the risk and burden of existing disease.

Magnetic resonance imaging (MRI) and 18F–fluorodeoxyglucose (FDG) positron emission tomography/computed tomography (PET/CT) are vital for the initial staging, therapeutic approach [10], and response assessment to treatment [11]. Studies in the past have explored the advanced functional imaging parameters obtained during MRI, namely, diffusion-weighted images (DWI) to identify tumors and further use the apparent diffusion coefficient (ADC) quantitative parameter from these DW images to assess response [12–16]. There is still a broad disparity regarding the selection of regions to measure ADC, as it argued that the selected part might not be representative of tumors exhibiting heterogeneity.

Radiomics is a growing arena of scientific research that uses imaging sets of high dimensional features, extracted from the normal acquired cross-sectional images and yields information that semantic analysis otherwise fails to acquire. The cystic and necrotic areas within the volume of the tumor that are representative of tumoral heterogeneity, and behavior that marks aggressiveness and hence outcome, are captured by radiomics [17,18]. This branch of science exploits mathematical modeling to dig quantitative features from medical images in order to gain predictive models that provide insight into treatment prognosis and survival [19–21], with preliminary studies [22–28] expressing a multitude of clinical outcomes by exploring radiomics. The present study endeavored to increase our existing knowledge regarding the role of functional imaging using diffusion-weighted derived quantitative parameters, namely the apparent diffusion coefficient (ADC) and the augmented role of radiomics using supervised classification algorithms by machine learning in the prediction of clinical outcomes, namely the FIGO stage, lymph node status, metastasis, and development of recurrence in uterine cervical cancer patients.

## 2. Methods

### 2.1. Patient Cohort with Treatment Characteristics

After obtaining approval from the institutional review board, this retrospective study was carried out between January 2016 and January 2017, in our institute, on patients who were referred to our hospital for a pelvic MR examination for the evaluation of histopathologically diagnosed uterine cervical cancer, fulfilled criteria for upfront treatment with chemoradiation, and were not surgical candidates. Eighty-three patients with cervical cancer of variable stage (FIGO IB2-IVA) were enrolled, details of which are outlined in the Supplementary File. The CPRS (Computerized patient record system—hospital information system) was reviewed to evaluate the patient's age, presence of para-aortic lymph nodes, development of distant metastasis and recurrence. Nodal recurrences were documented as pelvic or para-aortic. Metastasis was similarly recorded as to lung or other sites. The administration of radiotherapy may be delivered as external pelvic beam RT (EBRT), followed by brachytherapy or interstitial needle devices. The study cohort included patients who were given upfront EBRT in a total dose of 45 Grays in 25 fractions. An MR examination after the conventional EBRT was conducted to ascertain the status of any residual disease and facilitate the further decision for brachytherapy. As per the

institutional protocol, CT-based image-guided brachytherapy was performed. The ICRT dose was 7.5 Grays in 3–5 fractions, and the MUPIT dose was 20–25 Grays in 4–5 fractions.

### 2.2. Magnetic Resonance Imaging Technique

The standard non-contrast MRI of the pelvis was performed using the Siemens Avanto Magnetom 1.5 T MR Scanner. All patients were imaged in supine position using a pelvic body coil. Conventional and diffusion weighted (DW) MRI studies were conducted before the initiation and post completion of chemoradiation treatment. All of the patients underwent DWIs by using a multisection spin echo single shot echo planar imaging (EPI) sequence with b values of 0, 400, and 800 s/mm$^2$. An average of 15 sections was obtained in the axial plane covering the area of interest. Imaging parameters were: TR/TE of 10,000/108 ms, FOV of 40 × 40 cm, and acquisition matrix of 256 × 256 and section thickness of 5 mm with an intersection gap of 1–2 mm.

### 2.3. Conventional Image Analysis

A radiologist (XX), with more than six years of training in pelvic MR imaging, autonomously evaluated the Diffusion images and corresponding ADC maps, with an awareness of the fact that patients had cervical carcinoma but blinded to the final clinical outcome. Under the supervision of a board-certified radiologist with 25 years of experience in treating genitourinary cancers, quantitative DWI analysis of the tumor was performed, based on a freehand-drawn region of interest (ROI) on the ADC map [29] that showed restricted diffusion (i.e., high signal on b800 DWI and low signal on the ADC map); mean ADCs were recorded. The labeling of ROI was independently checked by a senior radiation oncologist (XX) with more than 25 years of experience in treating genitourinary cancers. If any, the disagreement was resolved by two senior radiologists (XX) with more than 40 years of experience. The ADC measured in initial baseline imaging was coded as ADCpre, and follow-up imaging post-treatment completion was coded as ADCpost, with a separate calculation for change referred to as ADCchange. Regarding lymph nodes, a positive node was defined by a short-axis diameter > 8 mm [30]. When in doubt, positive cytology was considered as the gold standard for the diagnosis of malignancy, both for pelvic and distal recurrence and distant metastasis.

### 2.4. Image Segmentation and Feature Extraction

All segmentations of the tumor were performed by a radiologist (XX) using the 3D Slicer software produced by Slicer [31] (http://www.slicer.org/, accessed on 12 December 2019). To minimize the errors involved in cropping from intra-operator and inter-operator variability, arising as a result of manual segmentation, a semi-automatic segmentation process was adopted [32]. This was further encouraged by the use of the Grow cut algorithm [33]. The task of 3D segmentation was aimed at culminating a volume of interest (VOI). The VOI consisted of regions of interest (ROIs) that were manually segmented along the tumor contour on each transverse section concerning T2 weighted images in the transverse axial planes. To remove any potential bias, the same radiologist re-segmented the VOIs, blinded to the previous task of segmentation, approximately two months after the first segmentation process. Finally, all segmentations were validated by a senior radiologist (XXX) who has 40 years of experience. A representative ROI and VOI definition are shown in the Supplementary File. Post-segmentation of images, using a semi-automated algorithm, 851 radiomics features were extracted using PyRadiomics [34], an open-source software package. The details of radiomic features in concordance with previous literature [35–37] are provided in the Supplementary File as text.

### 2.5. Radiomic Feature Selection

The Pearson correlation of radiomic features with clinical prognostications was used for selecting the features, and this was calculated using the stats v3.5.1 package in R v3.5.1 Features passing through any of the following correlation cut-off criteria were se-

lected for the model building: (i) correlation coefficient with Recurrence > 0.35 [cor($f_i$, Recurrence) > 0.35], (ii) correlation coefficient with Stage > 0.35 [cor($f_i$, Stage) > 0.35], (iii) correlation coefficient with Lymph Node > 0.49 [cor($f_i$, Lymph Node) > 0.49], (iv) correlation coefficient with Metastasis > 0.40 [cor($f_i$, Metastasis) > 0.40] (details of radiomics features are provided in the Supplementary File).

*2.6. Model Building*

After feature selection, we used multiple modeling algorithms to predict the following clinical outcomes: (i) recurrence, (ii) distant metastasis, (iii) lymph node metastasis, and (iv) FIGO stage. Various models with leave-out-one cross-validation and hyper-parameter tuning were trained using different classification algorithms using the caret v6.0-86 package for the R statistical software package. To find the significance of radiomics features over ADC parameters, each model was trained using different sets of features as follows: (i) 24 Radiomics features; (ii) 24 Radiomics features, preADC, postADC and change ADC (postADC—preADC); (iii) 24 Radiomics features, and preADC; (iv) 24 Radiomics features, and changeADC; (v) preADC, postADC2, and changeADC.

*2.7. Statistical Analysis*

Cohen's Kappa and area under the curve (AUC) were used as the statistical parameters to establish the model efficiency, which was calculated using the pROC v1.16.2 and caret packages available in R (details in Supplementary File).

**3. Results**

*3.1. Patient Characteristics and Disease Outcome*

The study enrolled 52 patients who presented with variable FIGO stages, with an age range of 28–79 (Median = 53 years) and a median follow-up of 26.5 months (range: 7–76 months). Disease recurrence occurred in 12 patients (23%). Four patients (33%) had an isolated pelvic recurrence and eight (67%) a distant recurrence (omentum = 1, peritoneum = 2, supra-clavicular node= 2, paraaortic node = 3), with a median recurrence-free survival of 19 months (range: 5–60 months). Metastasis occurred in 15 patients (28%). Ten patients had distant metastases to the lung, which were proven histopathologically in all cases, and clinical follow up with symptoms of the development of a second primary lung carcinoma ruled out. The remaining five patients showed metastasis to the peritoneum ($n$ = 3), spine ($n$ = 1) and liver ($n$ = 1). PreADC had a median of $0.615 \times 10^{-3}$ mm$^2$/s (range: $0.615–1.400 \times 10^{-3}$ mm$^2$/s) with an arithmetic mean value of $0.889 \times 10^{-3}$ mm$^2$/s, and postADC had a median of $1.760 \times 10^{-3}$ mm$^2$/s (range: $0.656–1.620 \times 10^{-3}$ mm$^2$/s with an arithmetic mean value of $1.469 \times 10^{-3}$ mm$^2$/s. The details of clinical characteristics are depicted in Table 1. The range of the selected 24 radiomics features and the average values are included in the Supplementary File.

*3.2. Application of Machine Learning Classifiers Algorithms to Predict Clinical Outcomes*

A monotone multi-layer perceptron neural network model generated with 24 radiomics features and preADC predicted recurrence with an AUC of 0.80 and a Kappa value as 0.55, outperforming other combinations of radiomics features (preADC, postADC and changeADC). AUC and Kappa generated by using only ADC features (preADC, postADC, and changeADC) are 0.57 and 0.17, respectively, using fuzzy rules with a weight factor, which shows that the addition of radiomics features to ADC values improves the results. The use of the same set of radiomic features for the prediction of metastasis combined with preADC, postADC and changeADC, in addition to a neural network with feature extraction to predict the distant metastasis, returns an AUC of 0.84 and a Kappa value as 0.65, which outperforms other possible combinations of radiomics features (preADC, postADC and changeADC). AUC and Kappa values generated using only ADC features (preADC, postADC, and changeADC) are 0.67 and 0.27, respectively, using fuzzy rules with a weight factor.

**Table 1.** Clinical characteristics of patients included in study.

| Clinical Parameters | Total *N* = 52 (%) |
|---|---|
| Age range | 28–79 (Median = 53 years) |
| FIGO Stage | |
| IB2 | 3 (5.7%) |
| IIA | 8 (15.5%) |
| IIB | 16 (30.7%) |
| IIIA | 16 (30.7%) |
| IIIB | 4 (7.7%) |
| IVA | 3 (5.7%) |
| Clinical Outcomes/Variables | |
| Recurrence/No recurrence | 12/40 (23%/77%) |
| Distant Metastatic/Non metastatic | 15/37 (28%/72%) |
| Metastasis to Lung/Other sites | 5/10 (9%/19%) |
| Lymph node Present/Absent | 15/37 (28%/72%) |
| Paraaortic lymph node/Pelvic node | 2/13 (3.8%/25%) |
| Mean follow up | 29.9 months |
| Median follow up | 28.5 months |
| Mean recurrence interval | 18.5 months |

The prediction of stage with 24 radiomics features, preADC, postADC and changeADC, using the k-nearest neighbors (KNN) algorithm, yields an AUC of 0.71 and a Kappa value as 0.25, which outperforms other possible combinations of radiomics features (preADC, postADC and changeADC). The use of the Regularized Random Forest (method = RRF-global) for the same caused a drop in AUC to 0.51 but improved the Kappa value to 0.31. The AUC and Kappa values generated using only ADC features (preADC, postADC, and changeADC) are 0.71 and 0.25, respectively, using the k-nearest neighbors (KNN) algorithm with a weight factor. This result is inconsistent with our previous results for recurrence and metastasis, and we did not achieve any marginal increment in our AUC or Kappa with the integration of radiomics into functional MR parameters for the prediction of stage. Lastly, the prediction of Nodal metastasis using 24 radiomics features, preADC, postADC and changeADC, in addition to the Evolutionary Learning of Globally Optimal Trees (evtree) with feature extraction, returns an AUC of 0.75 and a Kappa value as 0.60, while the combination of ADC parameters using evtree provides an AUC of 0.64 and a Kappa value of 0.32 (Tables 2–5).

**Table 2.** Tabulated content of recurrence with relevant classifier algorithm and corresponding AUC with Kappa values.

| Output | Features | Model | Metric | AUC | Kappa |
|---|---|---|---|---|---|
| Recurrence | Radiomics | pcaNNet (Neural Networks with Feature Extraction) | Kappa + AUC | 0.77 | 0.53 |
| | Radiomics + ADC1 + ADC2 + Change ADC | svmLinearWeights (Linear Support Vector Machines with Class Weights) | Kappa + AUC | 0.76 | 0.49 |
| | Radiomics + ADC1 | Monmlp (Monotone Multi-Layer Perceptron Neural Network) | Kappa + AUC | 0.8 | 0.55 |
| | Radiomics + change ADC | RRFglobal (Regularized Random Forest) | Kappa | 0.74 | 0.5 |
| | Radiomics + change ADC | svmLinearWeights (Linear Support Vector Machines with Class Weights) | AUC | 0.77 | 0.48 |
| | ADC | FRBCS.W (Fuzzy Rules with Weight Factor) | Kappa + AUC | 0.57 | 0.17 |

**Table 3.** Tabulated content of metastasis with relevant classifier algorithm and corresponding AUC with Kappa values.

| Output | Features | Model | Metric | AUC | Kappa |
|---|---|---|---|---|---|
| Metastasis | Radiomics | svmLinearWeights (Linear Support Vector Machines with Class Weights) | Kappa + AUC | 0.76 | 0.5 |
| | Radiomics + ADC1 + ADC2 + Change ADC | pcaNNet (Neural Networks with Feature Extraction) | Kappa + AUC | 0.84 | 0.65 |
| | Radiomics + ADC1 | pcaNNet (Neural Networks with Feature Extraction) | Kappa + AUC | 0.79 | 0.59 |
| | Radiomics + change ADC | pcaNNet (Neural Networks with Feature Extraction) | Kappa + AUC | 0.73 | 0.46 |
| | ADC | Rocc (ROC-Based Classifier) | Kappa | 0.63 | 0.3 |
| | ADC | svmLinearWeights (Linear Support Vector Machines with Class Weights) | AUC | 0.67 | 0.27 |

**Table 4.** Tabulated content of stage with relevant classifier algorithm and corresponding AUC with Kappa values.

| Output | Features | Model | Metric | AUC | Kappa |
|---|---|---|---|---|---|
| Stage | Radiomics | RRFglobal (Regularized Random Forest) | Kappa | 0.51 | 0.31 |
| | Radiomics | Knn (k-Nearest Neighbors) | AUC | 0.71 | 0.25 |
| | Radiomics + ADC1 + ADC2 + Change ADC | Earth (Multivariate Adaptive Regression Spline) | Kappa | 0.64 | 0.3 |
| | Radiomics + ADC1 + ADC2 + Change ADC | Knn (k-Nearest Neighbors) | AUC | 0.71 | 0.25 |
| | Radiomics + ADC1 | Evtree (Tree Models from Genetic Algorithms) | Kappa | 0.63 | 0.33 |
| | Radiomics + ADC1 | Knn (k-Nearest Neighbors) | AUC | 0.71 | 0.25 |
| | Radiomics + change ADC | Earth (Multivariate Adaptive Regression Spline) | Kappa | 0.64 | 0.31 |
| | Radiomics + change ADC | Knn (k-Nearest Neighbors) | AUC | 0.71 | 0.25 |
| | ADC | RRFglobal (Regularized Random Forest) | Kappa | 0.57 | 0.19 |
| | ADC | LogitBoost | AUC | 0.66 | 0.06 |

**Table 5.** Tabulated content of lymph node with relevant classifier algorithm and corresponding AUC with Kappa values.

| Output | Features | Model | Metric | AUC | Kappa |
|---|---|---|---|---|---|
| Lymph Node | Radiomics | evtree (Tree Models from Genetic Algorithms) | Kappa + AUC | 0.75 | 0.6 |
| | Radiomics + ADC1 + ADC2 + Change ADC | evtree (Tree Models from Genetic Algorithms) | Kappa + AUC | 0.75 | 0.6 |
| | Radiomics + ADC1 | evtree (Tree Models from Genetic Algorithms) | Kappa + AUC | 0.75 | 0.6 |
| | Radiomics + change ADC | evtree (Tree Models from Genetic Algorithms) | Kappa + AUC | 0.75 | 0.6 |
| | ADC | evtree (Tree Models from Genetic Algorithms) | Kappa + AUC | 0.64 | 0.32 |

Figures 1–4 depict the correlation between different radiomics and ADC features with the two classes of Lymph Node (Absent and Present), Metastasis (Absent and Present), recurrence and clinical outcomes, respectively. The details of the interpretation of these heat maps are provided in the Supplementary File.

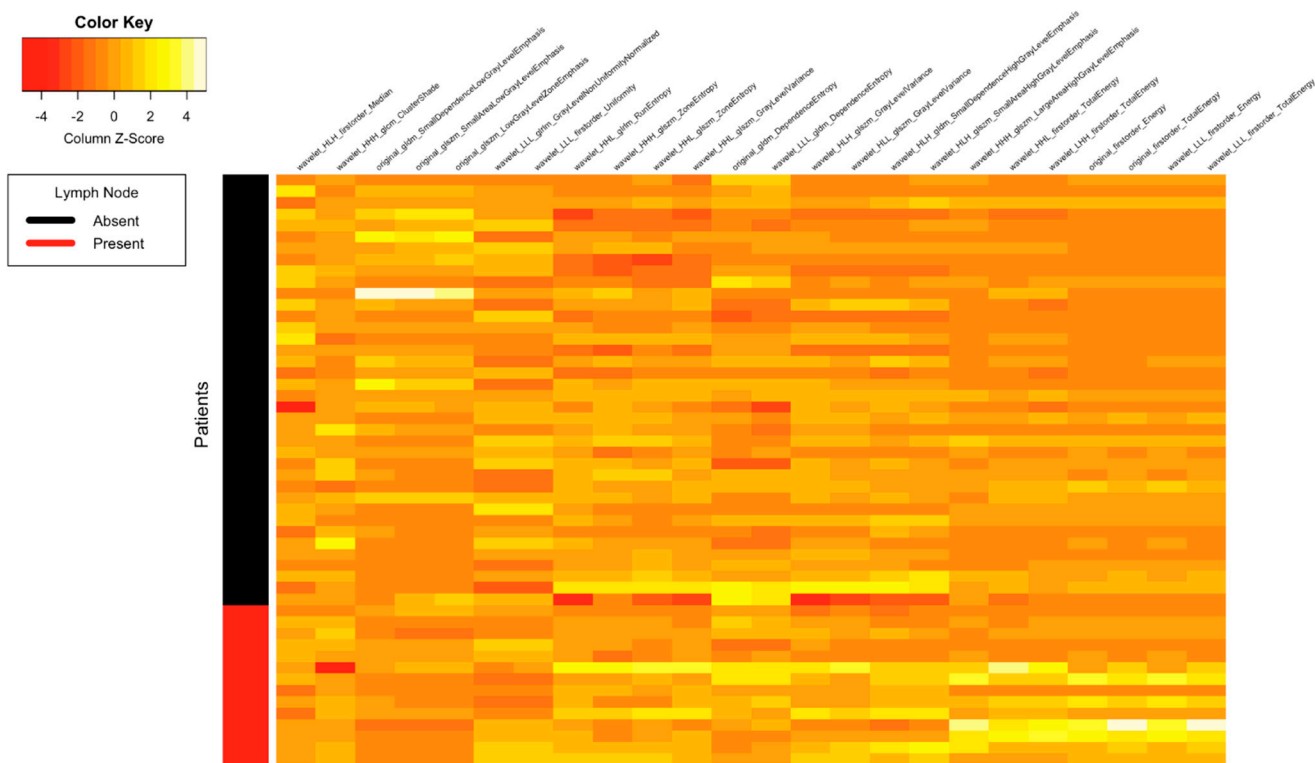

**Figure 1.** Heat map showing correlation between different radiomics and ADC features with the two classes of lymph node (absent and present).

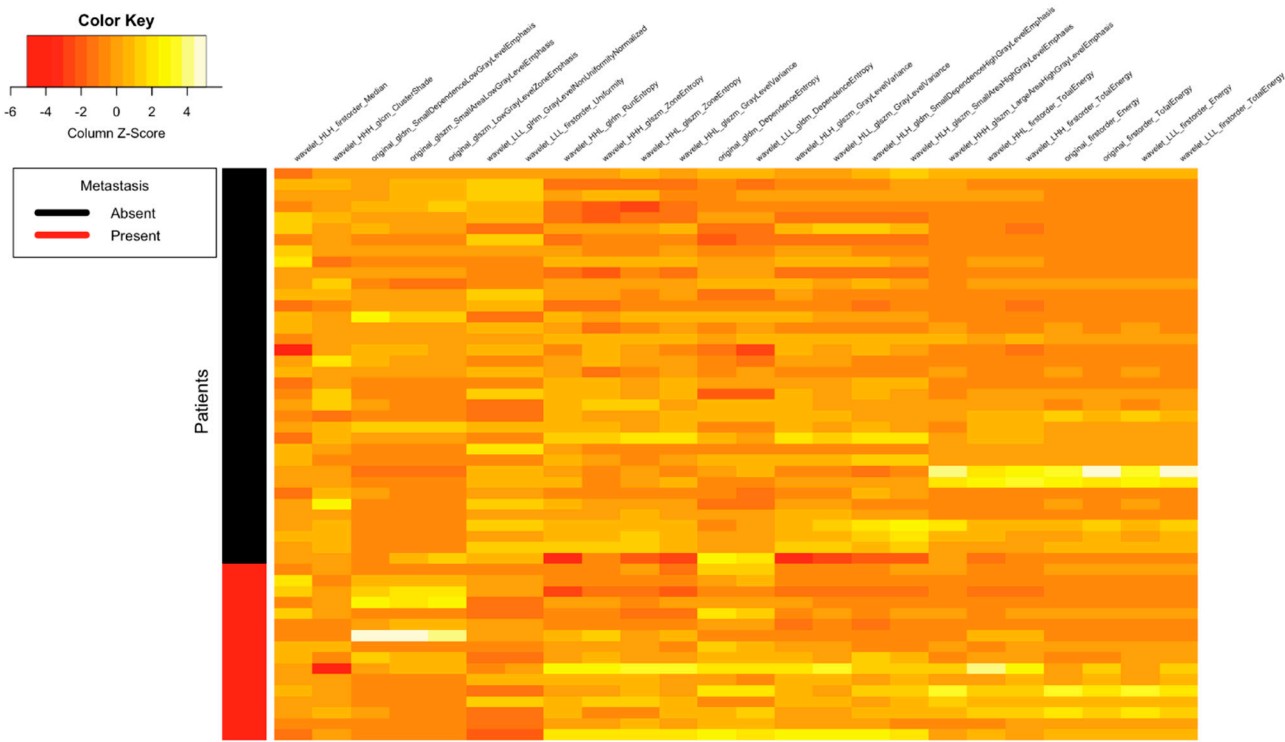

**Figure 2.** Heat map showing correlation between different radiomics and ADC features with the two classes of distant metastasis (absent and present).

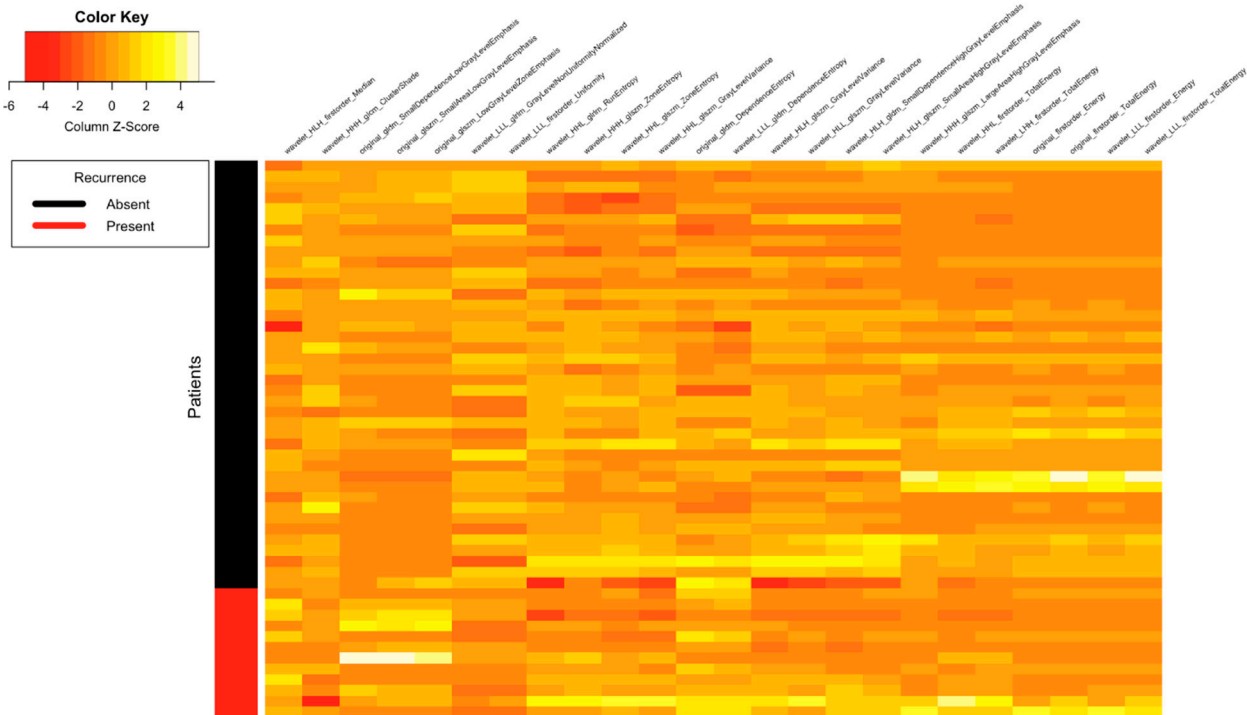

**Figure 3.** Heat map showing correlation between different radiomics and ADC features with the two classes of recurrence (absent and present).

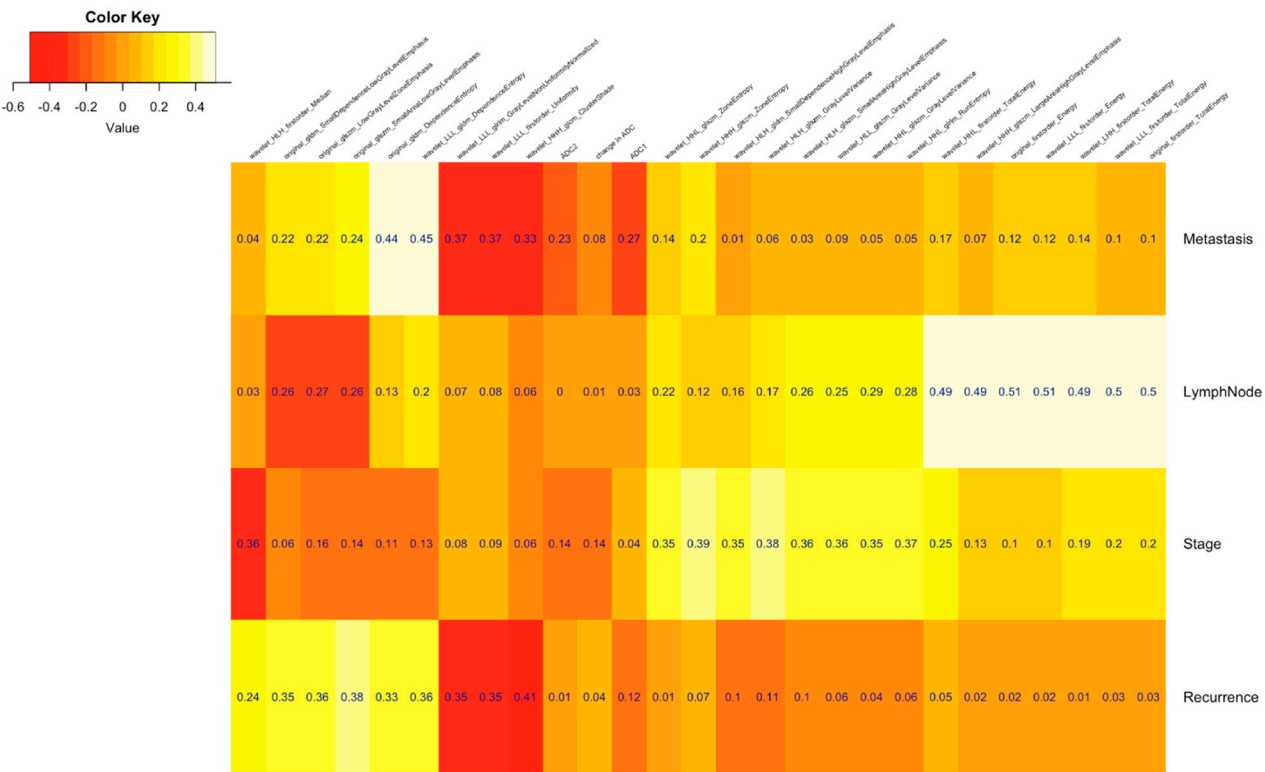

**Figure 4.** Heat map showing correlation between each feature with clinical outcomes desired in this study design.

## 4. Discussion

The results of this study show the original GLSZM and wavelet features to be correlated with clinical outcomes, with major contributions from the GLSZM features (SALGLE and LGLZE) and GLDM features (SDLGLE and DE) to recurrence and metastasis. This study documents the earliest correlation between coarseness, SALGLE, LGLZE, and difference entropy with clinical outcomes. Distant metastasis is a primary factor amounting to treatment failure despite good rates of local control in cervical cancer post-chemo-radiation, and has an incidence as high as twenty percent [38]. Recurrence is seen in a third of the patients and occurs shortly after treatment completion [39]. Up to forty percent of the patients with a positive para-aortic node have distant metastasis [40]. In our study, among three patients with paraaortic nodes, distant lung metastasis was seen in one patient.

A previous study of Mu W et al. found a statistically significant difference, with an AUC of 0.8, between the early and advanced stages by combining texture features derived from FDG-PET CT and SUV parameters using the SVM algorithm [41]. We achieved an AUC (0.71) using a higher number of patients with multiple classifiers to predict differences in stages, with the best AUC obtained with the k-nearest neighbors (KNN) algorithm. Due to differences in the acquisition and considering our granular approach of staggering the model output into four stages, rather than clubbing into early and advanced stages, a direct comparison of the study of Mu W et al. with this study is not possible. The majority of determinants of pelvic LN metastasis are assessed only in a post-operative setting, such as depth of stromal invasion and lymphovascular invasion [42]. Therefore, a robust parameter that could help in the pre-treatment prediction of LN metastasis would be desirable, as LN involvement is an independent prognostic marker for recurrence and overall survival [27,43,44]. While this study had an AUC of 0.864 for the prediction of LN involvement, we were able to achieve comparable levels with another study [45] (AUC = 0.75). Some radiomics features exhibited an escalating trend analogous to patients who developed either distant metastasis or recurrence versus patients with a lack of such events. Previous studies have shown an AUC of 0.747–0.85 to discriminate nodal metastasis by using radiomic features on functional ADC maps in cervical cancer [46,47]. We attained a similar AUC (0.75) using the evTree classifier on a similar number of patients. Furthermore, our study could also demonstrate that the combination of radiomic features with ADC parameters did not show any better performance than radiomics alone in LN assessment. This is in contrast to a study of Kan Y et al. that focused solely on the prediction of the nodal metastatic stage (AUC 0.75), with the integration of seven distinct clinical characteristics into the equation [48].

One of the wavelet features, HLH_GLSZM_small_area_low_grade_level_emphasis, was reported in an earlier study conducted on T2WI for the prediction of Disease-free survival (DFS) in uterine cervical cancer [49], using 18 radiomic features and lymphovascular space invasion (LVI) with contrast MRI, to obtain a Rad score for the prediction of the DFS. Another wavelet feature, LLL_glrlm_Gray_Level_NonUniformity_Normalized, was also found to be significant in the prediction of recurrence in uterine cervical cancer [50] and 5-year survival [51]. In concordance with the previous studies where GLCM Entropy is reported as a feature to predict Disease free survival [52], we found significant entropy in our results for GLDM, GLRLM and GLSZM. Unlike GLCM, which characterizes the local information on gray levels between pairs of voxels, GLRLM captures the coarseness and GLSZM quantifies the clusters of homogeneous intensity regions within the tumor. The feature energy has been reported to predict recurrence, with an AUC of 0.885, when derived from ADC maps in a previous study [50]. We found both first order and wavelet derivatives of energy to be useful in building our radiomics model, and, in combination with functional quantitative parameters, yielded an AUC of 0.84 in our study for the prediction of distant metastasis. A worse clinical outcome was associated with elevated values of these parameters, asserting the fact that a poor prognosis is exhibited by more heterogeneous tumors.

Most studies performed in the past have focused on the standardized uptake value (SUV) derived from PET CT imaging, with few exploring ADC maps on MR imaging as predictors of treatment response overall survival and lymph node involvement [53], and proved the superiority of radiomic features, in comparison to SUV, in the assessment of clinical outcome and as a descriptor of tumor heterogeneity [54]. Changes in ADC have been reported to be a promising imaging biomarker for early radiation response in prostate cancer [55]. Few studies have revealed the role of ADC as a predictor of recurrence [14,56]. In our study, pretreatment ADC (preADC) values did not show considerable performance as compared to radiomics features. One likely explanation for the above observation could be attributable to wide differences in methodology, where a certain ROI was drawn manually on ADC maps excluding the areas of necrosis. On a related note, the radiomics features achieved VOI in a semi-automatic fashion through the use of a grow cut feature that encompassed the whole volume of the tumor. This potentially captures the tumor heterogeneity, including the areas of necrosis, which are believed to steer the clinical outcome and response, even in uterine cervical cancer. The analysis of conventional methods of tumor assessment using diffusion sequences and ADC maps was prone to errors because ADC values were consequential from manually drawn ROIs. There are always chances of human errors while measuring the same, and the measurement also suffers from a certain degree of inter observer variability. Further restrictions in terms of efficacy occur in the assessment of the mean ADC change within the ROI of the tumor on a single ADC map image, which pertains to tumor heterogeneity in the post treatment response [57]. Few studies show the jarring results concerning the prognostic value of ADC in cervical cancer, some documenting poorer prognosis with a lower ADC value [58]. The VOI delineated on the T2W image could have been extrapolated on the ADC maps to extract the radiomic features on ADC images, as was the case in previous studies [59], but the additional normalization and binning involved was beyond the scope of this study. Undeniably, a better understanding of the core spatial heterogeneity could be offered by the VOI delineation and analysis of ADC imaging; however, the above factors, with their resource and time constraints, prevented us from exploring this particular aspect, thus limiting its usefulness in terms of routine clinical practice [60]. Further ADC values show heterogeneity between various MR scanners [61], which would be an obstacle in terms of standardization.

The monocentric and retrospective nature of studies are the limitations that have been repeated in most of the radiomics studies of uterine cervical cancers. Despite the use of cross-validation, a clear limitation is the fact that there was no validation dataset (Study cohort included only 52 patients overall, and hence, a split would not have made statistical sense). Being monocentric, however, the study erases problems that arise due to variations in acquisition and image reconstruction parameters, which have negatively affected analysis [62]. The study cohort, though small, included a large number of features (more than 800), but training and testing was performed using the Pearson correlation, and statistical significance was corrected through multiple rounds of testing with the application of leave-one-out cross validation and hyper parameter tuning in order to avoid both overfitting and false-discovery. As has been pointed out previously, most radiomics studies encompass higher numbers of features than patients, likely culminating in a high risk of false-positives [63]. Since the clinical variables rather than the outcomes were dichotomized, the supervised analysis conducted aimed towards the creation of a good union of these variables with the help of statistical learning models. In pursuit of this, we chalked out the most efficient radiomic features that could predict clinical outcomes.

## 5. Conclusions

Devising a robust and noninvasive strategy to assess tumor heterogeneity, prior to treatment, might have an insightful impact on the management of individual cancer patients through the early prediction of treatment outcomes with the prospect of modifying therapy as part of the precision medicine exemplar. We tried to devise a machine learning

prediction method for clinical outcomes by exploiting the multi classifier system. We further aim to validate our findings in larger cohorts, both within our centre and in multi-institutional centers. Our results revealed an incremental role of radiomics in functional MR imaging that deploys ADC values for the prediction of recurrence and distant metastasis. The model will be particularly helpful for low to middle income nations that have an increasing incidence of cervical cancer as our study does not incorporate the use of FDG, PET, CT and Contrast MR imaging, the availability and interpretation of which are limited in resource-constrained low resource nations.

**Supplementary Materials:** The following are available online at https://www.mdpi.com/article/10 .3390/tomography7030031/s1, Figure S1: Heat map showing correlation between different radiomics and ADC features with Stages, Figure S2: (A–D) Line plot is used to represent the variation in the efficiency metrics of the model using different sets of features and modeling algorithms. The above figure shows our various classifiers used to predict clinical outcomes with Kappa plotted on a scale of 0–1, Figure S3: (A–D) Line plot is used to represent the variation in the efficiency metrics of the model using different sets of features and modeling algorithms. The above figure shows our various classifiers used to predict clinical outcomes with AUC plotted on a scale of 0–1, Figure S4: T2 WI and diffusion imaging showing representative method of ADC calculation, Figure S5: Segmentation process using Slicer-3d software and VOI delineation for radiomics feature extraction, Table S1: Using Pearson coefficient method of cut-off, the final set of features has 24 radiomics features listed, which are: two first-order features; two GLDM gldm features; two GLSZM features and 18 wavelet features ((i) six first-order features; (ii) two GLDM features; (iii) seven GLSZM features; (iv) one GLCM feature; (v) two GLRLM feature), Table S2: The baseline and follow up ADC values in the various groups with percentage change in ADC. There is an obvious change in ADC in the recurrence and metastatic groups. The nodal positive and negative groups did not show much of change.

**Author Contributions:** Conceptualization, A.J., H.P., M.M., A.M., S.P. (Sunil Puri) and A.C.; Data curation, A.J., A.G., S.M. and S.P. (Sunil Pasricha); Investigation, A.J., A.G., H.P., M.M., S.M. and S.P. (Sunil Pasricha); Methodology, A.J. and A.G.; Project administration, M.M., S.M., S.P. (Sunil Pasricha), A.M., S.P. (Sunil Puri) and A.C.; Resources, H.P. and S.M.; Software, A.G. and A.M.; Supervision, A.M., S.P. (Sunil Puri) and A.C.; Validation, A.J., A.G. and A.M.; Visualization, S.P. (Sunil Pasricha); Writing—original draft, A.J.; Writing—review and editing, H.P. and M.M. All authors have read and agreed to the published version of the manuscript.

**Funding:** This research received no external funding.

**Institutional Review Board Statement:** IRB (ethical) approval was taken. Nos.RGCIRC/IRB-BHR/22/2020.

**Informed Consent Statement:** Patient consent was waived off with receipt of IRB approval.

**Data Availability Statement:** All data generated and analyzed during this study are included in this published article (and its Supplementary Information Files).

**Conflicts of Interest:** The authors declare no conflict of interest.

## Abbreviations

| | |
|---|---|
| MRI | Magnetic resonance imaging; |
| 18-FDG PET CT | 18F–fluorodeoxyglucose (FDG) positron emission tomography /computed tomography; |
| FIGO | International Federation of Gynecology and Obstetrics; |
| ADC | Apparent diffusion coefficient; |
| ICRT | Intracavitary brachytherapy; |
| FNAC | Fine needle aspiration cytology; |
| RFS | Recurrence-free survival; |
| VOI | Volumes of interest; |

| ROI | Region of interest; |
|-----|---------------------|
| GLCM | Gray level co-occurrence matrix; |
| GLRLM | Gray level run length matrix; |
| GLSZM | Gray level size zone matrix; |
| GLDM | Gray-level dependence matrix. |

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
