# Peer review of "Combination of Radiomics and Machine Learning with Diffusion-Weighted MR Imaging for Clinical Outcome Prognostication in Cervical Cancer"

_tomography, doi:10.3390/tomography7030031_

Round 1

Reviewer 1 Report

Jajodia et al. described the Combination of Radiomics and Machine Learning with 2 Diffusion-Weighted MR Imaging for Clinical Outcome 3 Prognostication in Cervical Cancer.

The topic is not new but methods have been well detailed.

Authors already acknowledged among the study limitations that the algorithms need external validation in order to be adopted in clinical practice.

Furthermore they reported only AUC as metric; also reporting overall accuracy, sensitivity and recall would have helped explaining better the performance of the models. 

The methods are correct but the sample size is not large enough.
Nevertheless, the manuscript is interesting.

Reviewer 2 Report

I have no substantial remarks on this well written manuscript, aimed to explore the potential of radiomics (alone and in combination with quantitative DWI) for predicting clinical outcome and prognosis of cervical cancer. The introduction is balanced (introducing the clinical and radiological background of the study, as well as the potential for radiomics to expand the current knowledge of the study topic), the research methodology is rigorous and well explained, the results follow from the data, and the discussion is detailed and supported by the relevant literature. Also tables and figures (including heat maps) illustrate the main study findings well and in a clear manner, and the overall manuscript text is quite easily readable despite the complexity of the topic.